# Balance Cell Apoptosis and Pyroptosis of Caspase-3-Activating Chemotherapy for Better Antitumor Therapy

**DOI:** 10.3390/cancers15010026

**Published:** 2022-12-21

**Authors:** Lingjiao Li, Shengmei Wang, Wenhu Zhou

**Affiliations:** 1Xiangya School of Pharmaceutical Sciences, Central South University, Changsha 410013, China; 2School of Pharmacy, Hunan University of Chinese Medicine, Changsha 410208, China; 3Academician Workstation, Changsha Medical University, Changsha 410219, China

**Keywords:** chemotherapy, therapeutic index, GSDME, targeting, cell death

## Abstract

**Simple Summary:**

Chemotherapy has been widely used in clinic to treat various types of tumors, although severe side effects have been the most critical limitation of this treatment modality. For a long time, it has been believed that toxicity derives from the apoptosis of normal cells induced by the use of chemotherapeutic drugs due to their off-target biodistribution. In 2017, a breakthrough finding by Shao Feng’s group showed that the side effects were related to pyroptosis caused by chemotherapeutic-drug-induced GSDME activation, and, interestingly, pyroptosis shares the same upstream signaling molecule with apoptosis, i.e., the caspase-3 activation. From then on, great research attention has been paid to chemotherapy-induced pyroptosis. In reality, pyroptosis is a “double-edged sword”, causing side effects in normal tissue, but also being able to promote antitumor effects, owing to its regulation of antitumor immunity. Therefore, rationally balancing the cell apoptosis and pyroptosis of caspase-3-activating chemotherapy is critically important for better antitumor therapy. This critical review aims to summarize recent progress in the field, focusing on how to balance cell apoptosis and pyroptosis for better tumor chemotherapy.

**Abstract:**

Chemotherapy is a standard treatment modality in clinic that exerts an antitumor effect via the activation of the caspase-3 pathway, inducing cell death. While a number of chemotherapeutic drugs have been developed to combat various types of tumors, severe side effects have been their common limitation, due to the nonspecific drug biodistribution, bringing significant pain to cancer patients. Recently, scientists found that, besides apoptosis, chemotherapy could also cause cell pyroptosis, both of which have great influence on the therapeutic index. For example, cell apoptosis is, generally, regarded as the main mechanism of killing tumor cells, while cell pyroptosis in tumors promotes treatment efficacy, but in normal tissue results in toxicity. Therefore, significant research efforts have been paid to exploring the rational modulation mode of cell death induced by chemotherapy. This critical review aims to summarize recent progress in the field, focusing on how to balance cell apoptosis and pyroptosis for better tumor chemotherapy. We first reviewed the mechanisms of chemotherapy-induced cell apoptosis and pyroptosis, in which the activated caspase-3 is the key signaling molecule for regulating both types of cell deaths. Then, we systematically discussed the rationale and methods of switching apoptosis to pyroptosis for enhanced antitumor efficacy, as well as the blockage of pyroptosis to decrease side effects. To balance cell pyroptosis in tumor and normal tissues, the level of GSDME expression and tumor-targeting drug delivery are two important factors. Finally, we proposed potential future research directions, which may provide guidance for researchers in the field.

## 1. Introduction

Cells are the basic units of structure and function of living organisms. The growth and development of the body are often accompanied by the renewal and death of crowds of cells, with the death of aging and damaged cells being as important as the survival of cells. Cell death requires genetically controlled programmed cell death (PCD) signals, such as apoptosis, pyroptosis, necroptosis, necrosis, and ferroptosis, among which apoptosis has been considered to be the main form of cell death [1,2,3]. In addition, apoptosis is the vital way for chemotherapeutic drugs to exert their efficacy [4,5,6]. Accompanied by a curative effect, however, a variety of toxic side effects has been observed during treatments [4]. Since apoptosis is often as silent as the fall of autumn leaves, the mild apoptosis of normal cells should not cause serious toxic side effects. Therefore, researchers have wondered whether there are other ways for chemotherapy drugs to induce death other than through apoptosis.

Pyroptosis, a new “star” death mode, is an inflammatory death mode different from apoptosis, with the characteristics of cell swelling and liberating cellular contents, such as inflammatory factors and immunogenic molecules, and it can be activated by the canonical inflammasome caspase-1-gasdermin-D (GSDMD) or nonclassical lipopolysaccharide (LPS)-caspase-4,5,11-GSDMD pathways [7,8,9]. The activation of GSDMD is mainly used to combat microbial infections and other related conditions [7], with other gasdermin family members having also been shown to induce pyroptosis [9,10]. Therefore, the study of molecules has the capacity to dig deeper into the properties of pyroptosis and advance the prevention and treatment of cancer and related diseases. Recently, investigators have focused on another member of this family—GSDME. Anecdotal evidence has suggested that under the stimulation of TNF or chemotherapeutic drugs, caspase-3 is activated and cleaves GSDME into pore-forming aminoterminal fragments, which transfers to the membrane to oligomerize into pores, releasing inflammatory factors and immunogenic molecules and, finally, shifting the mode of cell death from apoptosis to pyroptosis [11,12]. For a long time, caspase-3 was regarded as the only marker of apoptosis. This study could change people’s cognition and provide new insights into the prevention of drug treatment-related side effects.

GSDME is not only associated with drug toxicity and side effects, but also involved in tumor treatment. Studies have reported that when cancer cells are induced to undergo pyroptosis through specific means, immunogenic substance damage-associated molecular patterns (DMAPs), such as high-mobility group box 1 (HMGB1), calreticulin, and ATP, can be released to turn “cold” tumors into “hot” tumors, so as to improve immune efficacy, which is a good anticancer strategy [12]. What factors determine whether apoptosis or pyroptosis occur? Shao Feng’s research group constructed a HeLa cell line (carrying green fluorescent protein) that stably expressed GSDME, and used flow cytometry to sort through cells with high and low GSDME expressions separately [11]. The results showed that cells with a high GSDME expression directly underwent pyroptosis and the low expression cells underwent apoptosis at first and then pyroptosis, while the cells without GSDME expression only underwent apoptosis. Therefore, these results indicated that the expression level of GSDME determines the mode of death after caspse-3 activation.

However, there is no study providing criteria for the definition of high and low levels of GSDME expression. If some researchers were to study this in depth and obtain a threshold, this could advance clinically relevant assays of GSDME. Unfortunately, GSDME tends to be methylated to appear as gene silencing in cancer cells [13]. Recently, several studies reported that the use of methyltransferase inhibitor decitabine (DAC) could reverse the expression of GSDME and, thus, improve the sensitivity of tumor cells to chemotherapeutic drugs (doxorubicin and cisplatin), providing new insights into the mechanism of chemotherapeutic drug resistance [14,15]. Furthermore, clinical evidence suggests that the application of doxorubicin leads to cardiotoxicity, which may be associated with high GSDME expression in cardiomyocytes [15]. Therefore, one of the key points of future applications is how to effectively use the function of GSDME to controllably switch these two cell death modes, inducing pyroptosis to reduce drug resistance in tumors, whilst, in the meantime, attenuating pyroptosis in normal tissues to avoid their damage.

Considering the importance of chemotherapy in clinic, this is a thriving tumor therapy field, and rationally balancing chemotherapy-induced apoptosis and pyroptosis is a promising strategy towards improving the therapeutic index. Given the significant advancements in the field, we believe it would be the right time to summarize the recent progress and speculate future directions. In this critical review, we summarized the basic mechanisms of cell apoptosis and pyroptosis, in which caspase-3 plays critical roles in both modes of cell death. We systematically illustrated pyroptosis as a “double-edged sword” for enhanced antitumor activity and increased side effects (Figure 1A,B), emphasizing the importance of regulating pyroptosis. Subsequently, the methods to switch from apoptosis to pyroptosis in tumors and block pyroptosis in normal organs were reviewed (Figure 1C,D). Finally, the current challenges of balancing cell apoptosis and pyroptosis were introduced (Figure 1E), and future perspectives were speculated for researchers in related fields.

## 2. Cell Apoptosis: A Classical Mechanism for Chemotherapy

### 2.1. Overview of Cell Apoptosis

Apoptosis has been a proposed concept for more than a century, and can be traced back to 1842, when Vogt discovered a cell death phenomenon different from cell necrosis when studying the development of tadpoles. In 1885, Flemming again discovered the cell death phenomenon when studying ovarian follicular cells, and he, for the first time, pointed out that this cell death was a part of the physiological function of the organism, named chromatolysis death. However, due to limitations in theory and technology at the time, this phenomenon was overlooked for a long time. In 1972, Kerr et al. introduced the concept of apoptosis into the biological world, becoming the real beginning of the exploration of apoptosis [16]. They found that apoptosis mainly manifested during chromatin condensation, cell shrinkage and the formation of apoptotic bodies, laying the foundation for the subsequent research on apoptosis conducted by researchers, from then on beginning to study apoptosis-related molecular mechanisms. For example, Vaux et al. found that B cell lymphoma/leukemia 2 (Bcl-2) could inhibit apoptosis [17]. In addition, many genes related to apoptosis have been deeply studied.

During the period from the 1990s to the early 2000s, there was an important understanding of the mechanisms of apoptosis, mainly including intrinsic and extrinsic pathways: (1) The extrinsic pathway (Figure 2A) is initiated by members of the death receptor family, such as Fas (CD95), TRAIL-R1 (DR4), and TRAIL-R2 (DR5), which bind to their corresponding ligands [18]; Fas-associated death domain (FADD) and cysteine proteases such as caspase-8 are recruited to form a death-inducing signaling complex (DISC) [19]. This complex can activate downstream caspase-3, which eventually leads to apoptosis. (2) The intrinsic pathway (Figure 2B) is triggered by the mitochondrial outer membrane permeabilization (MOMP) mediated by the Bcl-2 family under conditions such as DNA damage and oxidative stress [20]. Subsequently, cytochrome c is released into the cytoplasm, where it binds with apoptotic protease-activating factor-1 (APAF-1) to form apoptotic bodies, which activate caspase-9 and further activate downstream caspase-3, ultimately leading to apoptosis. Above all, caspase activation is critical for regulating the intrinsic and extrinsic pathways of apoptosis, and since caspase-3 is located at the end of the caspase cascade and activated by both intrinsic and extrinsic pathways, it is considered to be the most important protein in apoptosis.

### 2.2. Chemotherapy to Cause Cell Apoptosis for Tumor Therapy

Apoptosis plays an important role in the development of various pathological states. For example, the excessive apoptosis of normal cells can induce neurodegenerative diseases, HIV infections and autoimmune diseases, while a limited apoptosis of tumor cells can induce tumors. Therefore, in recent years, researchers have been working to develop drugs that target apoptosis to treat various diseases.

The main pathogenesis of cancer may be due to the loss of the ability of cells to accept apoptosis-induced death, causing cell proliferation to be out of control. As described above, bcl-2 is an apoptosis-related gene with cell growth-promoting properties. The identification of the bcl-2 oncogene at the t(14;18) translocation chromosomal breakpoint in a human leukemia line was the first evidence to suggest that apoptotic mechanisms are involved in tumorigenesis [21]. Later, the amplification of the *Bcl-2 gene* was also found in follicular lymphoma [22]. In addition, approximately 50% of cancer cells were found to show markedly elevated bcl-2 expression. However, unlike other oncogenes, bcl-2 was not shown to disrupt normal proliferation control, but promoted cell survival by blocking programmed cell death. In addition, death receptors and their ligands are key players in the extrinsic pathway of apoptosis, and their abnormal expression is also a way for cancer to escape death signals [23]. Therefore, in the pathogenesis of tumors, the abnormal mechanism of apoptosis involves almost all aspects of the apoptosis signaling pathway. The above analysis showed that defects in the apoptosis pathway play a crucial role in carcinogenesis, and, therefore, targeting apoptosis is a useful therapeutic strategy. For example, the selective inhibition of Bcl-2 family members is an important tool for tumor therapy in targeting endogenous apoptotic pathways. Venetoclax (Bcl-2-selective inhibitor) is used to treat small lymphocytic lymphoma (SLL), which mainly promotes tumor cell apoptosis by inhibiting Bcl-2 and, ultimately, activating caspase-3 [24,25,26,27]. In targeting the exogenous apoptosis pathway, targeting death receptors and ligands is the main means of tumor treatment. For example, various antitumor drugs, such as doxorubicin, methotrexate, and cytarabine, can stimulate a Fas/Apo –1/CD95 interaction to induce tumor cell apoptosis. For a long time, caspase-3 was considered an important marker of cell apoptosis. This means that different stimulatory signals may initiate apoptosis through different pathways, but the final common pathway is the activation of caspase-3. Therefore, various clinical therapeutic strategies attempt to induce apoptosis by activating caspase-3. However, although the induction of tumor cell apoptosis is a common therapeutic strategy, multidrug resistance often occurs during the treatment, resulting in poor efficacy.

### 2.3. Antiapoptotic Pathways in Tumor Cells for Chemoresistance

Most cancer treatments aim to eliminate malignant cells by inducing apoptosis and other related cell deaths. However, with the increasing use of drugs, the antiapoptotic ability of tumor cells is gradually enhanced. Generally speaking, antiapoptotic pathways are mainly divided into the following mechanisms: (1) The disruption of the balance of proapoptotic and antiapoptotic proteins (Figure 3A). For example, the leukemia 2 (bcl-2) family are among the earliest studied genes associated with apoptosis. Bcl-2 plays a key role in the regulation of apoptosis, and is composed of proapoptotic and antiapoptotic proteins. When the balance between them is disrupted, this leads to an imbalance in apoptosis [28]. For example, myeloid cell leukemia 1 (McL-1), a member of the Bcl-2 family, is highly expressed in tumors, thereby blocking apoptosis, and has been found to be implicated in chemotherapy resistance [29,30]. Therefore, the imbalance between antiapoptotic protein and proapoptotic protein causes the apoptosis ability of cancer cells to be disordered, and, thus, resistant to various chemotherapy drugs. (2) Autophagy and cancer resistance of (Figure 3B) can maintain intracellular balance by removing dysfunctional organelles and removing cellular stress so as to prevent cell damage caused by chemotherapy drugs. Studies have shown that autophagy significantly inhibits the efficacy of a variety of anticancer drugs, protecting cancer cells from apoptosis and helping to acquire drug resistance [31]. (3) The existence of heat shock protein 90 (hsp90). The content of hsp90 in cancer cells (Figure 3C) is higher than that in normal cells. As a molecular chaperone, it assists the folding, stabilization, and maturation of different types of oncoproteins, thereby resisting apoptosis induced by chemotherapy drugs and promoting tumor proliferation, migration and metastasis. (4) The decreased surface expression or function of death receptor signaling (Figure 3D) can lead to the reduced apoptosis of tumor cells. For example, the downregulation of CD95 expression in neuroblastoma cells is related to its acquired drug resistance.

## 3. Caspase-3/GSDME Pathway-Mediated Cell Pyroptosis

### 3.1. Overview of Pyroptosis

Pyroptosis is a form of programmed cell death associated with inflammation. The specific phenotypes include cell swelling and rupture, with the release of inflammatory cytokines and highly immunogenic molecules, which can mediate the development of disease processes by activating immunity and creating an inflammatory microenvironment. Pyroptosis has been highly appreciated by researchers because of its close association with the development of many diseases, with studies involving pyroptosis having rapidly increased recently based on data from PubMed (Figure 4).

The discovery of pyroptosis took place decades ago. In 1992, Zychlinsky and other scientists observed pyroptosis for the first time in Shigella-flexneri-infected macrophages, which was only recognized as “apoptosis” at that time due to the theoretical and technological limitations [32]. It was not until 2001, when Brennan MA and Cookson BT published a paper, that the concept of “pyroptosis” was formally proposed [33]. In recent years, pyroptosis was redefined as a regulated cell death dependent on the formation of plasma membrane pores by the gasdermin family [34]. The gasdermin family, consisting of gasdermin-A (GSDMA), gasdermin-B (GSDMB), gasdermin-C (GSDMC), gasdermin-D(GSDMD), GSDME/DFNA5, and pejvakin (PJVK/DFNB59), was found to comprise proteins with perforating activity. All members of this family, except pejvakin, have an autoinhibitory structure, where the C-terminal domain masks the lipid-binding portion of the N-terminal structural domain [35]. Caspase-1/3/4/5/8/11 has been found to cleave gasdermin proteins, separating the C-terminal structural domain from the N-terminal structural domain, thus, mediating pyroptosis [7]. The activation of pyroptosis is classically induced in two ways [7], the first one being that the inflammasome composed of pathogen-related molecular patterns activates caspase-1 and then cleaves the effector protein GSDMD to liberate an active N-terminus domain, which oligomerizes into pores on the membrane, allowing water to enter and inflammatory cytokines to be released to induce pyroptosis (Figure 5A). The other way involves the activation of caspase-4, -5, and -11, directly induced by LPS in microorganisms such as bacteria to cleave GSDMD (Figure 5B). The identification of the structure and physiological function of GSDMD would provide researchers with a deeper understanding of pyroptosis and a more promising direction for the treatment of cancer and related diseases. However, GSDMD is mainly related to diseases such as infections, and only specific substances can induce the activity of GSDMD. In 2017, Shao Feng’s research group found that the toxicity and side effects of chemotherapeutic drugs may have a necessary relationship with GSDME, another member of the GSDM family, and they proposed a clear induction mechanism for GSDME. This discovery expanded the directions for the application of pyroptosis.

### 3.2. The Discovery of GSDME and Its Mechanism on Pyroptosis Induction

GSDME, also known as DFNA5 (deafness autosomal dominant nonsyndromic sensorineural 5), was initially thought to be a gene associated only with hearing function [36,37], but which was subsequently shown to act as a “tumor suppressor” [38]. For example, etoposide had no effect on a melanoma treatment when GSDME was removed [39]. Although GSDME does exert tumor-suppressive effects, no studies have been conducted to elucidate the specific mechanism of its tumor suppression. In recent years, GSDME was found to mediate pyroptosis, and its association with disease prevention and treatment has received considerable attention. In a published study, Shao Feng et al. noted that GSDME is similar to GSDMD protein, which can be cleaved with specific molecules to form an N-terminal fragment with pore-forming properties [11]. However, there are some differences between them: 1. Inducing factors, where GSDMD is mainly associated with microbial infection and danger signals, while GSDME is mainly caused by chemotherapeutic drugs, radiation, etc. 2. The existence of inconsistent cleavage sites in the linker motif, i.e., the GSDME cleavage site is “DMPD”, while for GSDMD it is “FLTD” [11]. 3. Due to the inconsistency of target proteins, studies reported that the upstream activating molecule of GSDME is caspase-3 instead of caspases −1, −4 −5, and −11 [11,40]. The discovery of GSDME not only expanded the mechanism and application of pyroptosis, but also added an important element to the cell death mode.

The physiological function and biochemical mechanism of GSDME still require elucidating, but its mechanism of inducing pyroptosis has been explored by many researchers. Similar to GSDMD, it mainly induces pyroptosis through three steps (Figure 6). First, the GSDME-N-terminal fragments liberate (Figure 6A). Studies have shown that the full-length GSDME protein does not have the ability to induce pyroptosis [11]. It contains an N-terminal domain and a C-terminal connected with a linker motif of approximately 30–100 amino acids, forming an autoinhibitory mechanism. When caspase-3 is activated, it cleaves a peptide segment consisting of four amino acids (DMPD) in a middle linker motif of GSDME to release the N-terminal with pore-forming activity. Then, the GSDME binds lipids in the cell membrane and oligomerizes to form functional pores [41] (Figure 6B). The pore-forming process of the GSDME-N contains multiple steps, including lipid binding, assembly, and forming functional pores. Studies have shown that the GSDME-N-terminus can bind to phospholipids on the membrane and oligomerize to form pores, which may be related to the positive charge of the GSDME-N-terminus, and the negative charge of the phospholipids; therefore, they can bind through an electrostatic interaction [11]. Notably, GSDME-N does not have strong affinity for every phospholipid. The phospholipid-binding assay verified that GSDME mainly binds to intracellular phospholipids (PSs) and phosphatidylinositol phospholipids (PIPs), rather than extracellular phospholipids to oligomerize into pores [11]. In addition, GSDME was shown to have little binding force to cardiolipin in mitochondrial membranes, but the liposome leakage experiments showed that it has a strong perforation ability in mitochondria, which indicates that GSDME can oligomerize perforation with only a very slight binding force to phospholipids. Next, cell membrane swelling and rupture (Figure 6C) studies have shown that the oligomerization of the gasdermin-N-terminus on the membrane mainly forms small pores with a diameter of 10–15 nm, and with the assistance of osmotic pressure difference, small inflammatory molecules such as interleukin-1β(IL-1β) and interleukin-18 (IL-18) are perforated and released outside the cell, attracting relevant immune cells to create a highly immune inflammatory microenvironment [9].

### 3.3. GSDME Signaling Pathways and Its Activation in Tumor Therapy

#### 3.3.1. GSDME-Related Signaling Pathways

GSDME is a newly tapped “star “molecule, and researchers have shown that it can be activated in two main ways, including caspase-3 and granzyme B cleavage (Figure 2C). In caspase-3–GSDME signaling pathways, apoptosis upstream proteins can promote the occurrence of pyroptosis, such as ROS-JNK/BAx-caspase-3-GSDME and NF-kB-cytochrome c-BAX-caspase-3-GSDME. In addition, scientists have pointed out that GSDME-N can also target mitochondria, promoting the release of cytochrome c to continuously activate apoptotic bodies and caspase-3, thereby establishing a self-amplifying feed-forward loop in the process of pyroptosis [42] (Figure 2D). In the granzyme B–GSDME signaling pathway, granzyme B released by natural killer cells (NK cells) can activate GSDME through directly cleaving the same site of caspase-3 or indirectly activating caspase-3 to activate GSDME [12] (Figure 2E). However, the detailed mechanism still remains to be fully understood.

#### 3.3.2. Activation of GSDME in Tumor Therapy

With the better understanding of the activation of GSDME, it has been found that the activation of GSDME can affect tumor therapy from both the efficacy and side effect aspects, which were illustrated in this section.

Growing evidence has demonstrated that GSDME can act as a tumor suppressor, and an increased GSDME expression in tumors can improve its antitumor efficacy. On the one hand, it has been demonstrated that GSDME can activate apoptosis. Studies have reported that GSDME can migrate to the mitochondrial membrane to oligomerize into pores after activation, which then induces a cytochrome c release and activates both endogenous and exogenous apoptosis [42]. On the other hand, GSDME induces pyroptosis to activate immunity [12]. The cleavage of GSDME can release cellular contents (DAMPs, chemokines, and cytokines) and promote a more inflammatory and immunogenic microenvironment, thus, acting as a tumor suppressor. As a result, inflammatory factors attract natural killer cells and CD8+ T cells to the tumor site and directly eliminate cancer cells, with NK cells able to release granzyme B to further induce pyroptosis. However, GSDME often has the characteristics of expression inhibition in the process of tumor occurrence and development, and genetic screening and microarray analyses showed a hypermethylation in the GSDME DNA promoter region [15]. Therefore, how to improve its expression in tumors has become a promising research direction, given the numerous advantages of GSDME cleavage in tumors.

Along with the discovery that GSDME is often highly expressed in normal tissues, such as the brain, heart, kidneys, etc., the other hot topic of GSDME research focuses on the drug-mediated activation of GSDME bringing indispensable side effects, such as nephrotoxicity caused by cisplatin and cardiotoxicity caused by doxorubicin. In addition, the emergence of toxic side effects often result in bad patient experiences, and even affect the treatment outcome. Therefore, how to reduce its expression in normal tissues to reduce toxic side effects is also a very meaningful topic of study. Overall, it is important to skillfully balance the cutting of GSDME to increase the efficacy, while circumventing the toxic side effects of chemotherapy.

## 4. Switching from Apoptosis to Pyroptosis for Better Antitumor Activity

In recent decades, chemotherapeutic drugs, whose target is apoptosis, have achieved significant advancements. However, some studies have shown that the resistance of tumor cells to apoptosis and the immunosuppression of tumor microenvironments can lead to the poor effect of tumor treatments [43,44]. Currently, cancer immunotherapy is the main treatment strategy, with the main approaches including chimeric antigen receptor T-cell (CAR-T cell) therapy and the use of immune checkpoint inhibitors. However, not all patients can be treated using these approaches, and up to 85% of cancer patients have innate or acquired resistance to immune checkpoint inhibitors. In addition, CAR-T cell therapy also carries significant risks. Therefore, new approaches to combat tumor cells are urgently needed to avoid apoptosis and enhance antitumor immunity. Recently, researchers found that GSDME-mediated pyroptosis can release various inflammatory factors and DAMPs, and stimulate a strong inflammatory response to regulate the tumor immune microenvironment, resulting in the activation of an antitumor immune response. Moreover, caspase-3 is a common effector of GSDME-mediated pyroptosis and apoptosis. Thus, promoting the transition from apoptosis to pyroptosis also seems to be a good cancer treatment. Next, we introduced the relationship between GSDME-mediated pyroptosis and antitumor immunity, as well as the methods to activate pyroptosis.

### 4.1. GSDME-Mediated Pyroptosis and Tumor Immunity

Tumor immunotherapy has recently captured people’s interest. Clinical studies have shown that immunotherapy is effective in the treatment of advanced tumors, allowing many cancer patients to prolong their survival time and even eventually recover [45,46]. It can inhibit and kill tumor cells by restarting and maintaining the tumor-immune cycle, so as to restore the normal antitumor immune response of the body. By restarting and maintaining the tumor-immune cycle, inhibiting and killing tumor cells, the body can restore a normal antitumor immune response. As we all know, the tumor immune microenvironment (TIME), which refers to the microenvironment related to immune cells, such as tumor-associated macrophages (TAMs), NK cells, myeloid-derived suppressor cells (MDSCs), and T cells in the tumor microenvironment, plays a crucial role in the process of tumor formation, and tumor cells can escape immune surveillance through the TIME. Therefore, changing the tumor microenvironment into an immune stimulatory state has an important role in tumor immunotherapy.

In recent years, several studies have shown that the chemotherapy-induced activation of GSDME can enhance antitumor immunity through forming an immunostimulatory state. For example, Judy Lieberman et al. found that GSDME overexpression significantly increased the number of CD8 cytotoxic T cells and NK cells in tumors, and that granzyme B sourced from these could directly cleave GSDME to mediate cancer cell pyroptosis, turning the TIME from “cold” to “hot”. As a result, the activated state of the TIME can effectively kill tumor cells [12]. In addition, Shao Feng et al. constructed a bio-orthogonal chemical system to control the antibody–drug coupling dissociation, the results of which showed that the pyroptosis of only a small fraction of cells could eliminate tumors. They found increased numbers of NK cells, T cells, and M1 macrophages, while finding decreased numbers of M2 macrophages, regulatory T cells, neutrophils, and MDSCs. Therefore, the functions of pyroptosis and cytotoxic lymphocytes in the TIME were mutually reinforcing, forming a positive feedback loop in anticancer immunity [47]. Furthermore, Zhou et al. found that GSDME induced antitumor immune cell infiltration through mediating pyroptosis, which enhanced cisplatin sensitivity [48]. All the above studies suggested that pyroptosis may play a vital antitumor role by activating the immune response.

It should also be noted that promoting tumor pyroptosis may not always be beneficial. For example, HMGB1 promotes the activation of caspase-1 and GSDMD in macrophages during tumor cell pyroptosis mediated through CAR-T/GSDME immunotherapy. A high release of IL-6 and IL-1β can induce cytokine release syndrome (CRS). Therefore, if we were to want to improve the tumor microenvironment by inducing pyroptosis, we may need to find the optimal degree of pyroptosis. As such, the induction of pyroptosis can improve the antitumor efficacy through multiple dimensions, while studies have shown that the occurrence of apoptosis or pyroptosis is regulated by the expression level of GSDME. However, the GSDME expression is sometimes limited because of promoter methylation. Therefore, improving the level of pyroptosis in tumor cells is particularly important.

### 4.2. Methods of Transformation Apoptosis into Pyroptosis

#### 4.2.1. Induction of Epigenetic Modifications

Tumors are a disease caused by the accumulation of genetic abnormalities due to mutations in oncogenes or/and tumor suppressor genes. In fact, carcinogenesis involves genetic and epigenetic changes. The latter includes chromosome remodeling, histone modifications, DNA methylation, and noncoding RNA regulation [49]. They regulate gene activity without changing the DNA sequence, and participate in the regulation of gene expression, playing a key role in cell proliferation and cell differentiation. However, their abnormal expression may induce the development of various malignant tumors and tumor drug resistances [50]. Therefore, targeting epigenetic alterations may inhibit tumorigenesis and drug resistance.

Histone deacetylases are highly expressed in tumor cells, and can reduce the histone acetylation and downregulate the expression of proapoptotic genes and tumor suppressor genes, which may turn out to be a new tumor treatment target. Recently, researchers identified the histone deacetylase inhibitor (HDACI) to be an important epigenetic regulator, which plays a decisive role in inducing pyroptosis by promoting the acetylation of histone and nonhistone substrates (Figure 7A). For example, Chen et al. designed a GSH-responsive nanogel (LD NP) coloaded with LAQ824 (a typical HDACI) and DOX, which can release the two drugs in the tumor microenvironment, activate caspase-3 to mediate pyroptosis induced by GSDME, and, thus, promote antitumor immunity [51].

When GSDME is highly expressed, chemotherapy drugs can activate caspase-3 to cleave GSDME, changing the cell death pathway from apoptosis to pyroptosis [52]. However, the epigenetic silencing and low level of GSDME in most tumors are mainly due to the methylation of the 5’-flanking region of GSDME [53,54,55]. When the GSDME gene is abnormally methylated, it causes changes in the chromatin structure, thus, inactivating transcription and silencing GSDME expression. Therefore, the demethylation of tumor cells with methyltransferase inhibitors may be a good strategy (Figure 7A). Fujikane et al. found that the upregulation of GSDME using the methyltransferase inhibitor DAC sensitized cancer cells to apoptosis or secondary necrosis induced by chemotherapy drugs [56]. In addition, there is evidence that cancer cells are pretreated with DAC to restore GSDME expression. Chemotherapeutic drugs can convert caspase-3-dependent apoptosis into pyroptosis through GSDME. However, due to the high expression of GSDME in normal cells, pyroptosis also occurs in normal cells after treatment with chemotherapeutic drugs, which may be the potential mechanism of toxicity and side effects of chemotherapy drugs [11,57]. Researchers often use nanotechnology to address this limitation. For example, Fan et al. combined DAC with chemotherapeutic nanodrugs to trigger the pyroptosis of tumor cells. They first pretreated tumor cells with DAC and then administered cisplatin-loaded tumor-targeting nanoliposomes (LipoDDPs), which activated the caspase-3 pathway in tumor cells and triggered pyroptosis [15].

Overall, GSDME expression levels determine the type of programmed cell death that occurs in cells activated with caspase-3, and cells with high levels of GSDME expression usually undergo pyroptosis after treatment with chemotherapeutic agents that activate apoptosis. These studies showed that epigenetic therapy cannot only reactivate silenced genes in tumor cells, but also restore exhausted T cells and increase immune cell infiltration. Therefore, epigenetic modification could gradually become a new research direction for inducing tumor cell pyroptosis.

#### 4.2.2. Autophagy Inhibition

Autophagy is a form of cell self-degradation, which degrades damaged cytoplasmic proteins or organelles through lysosomes, thereby realizing the metabolic needs of the cell itself and the renewal of some organelles [58,59]. Atg plays an important role in the process of autophagy. As early as 2008, Saitoh et al. found that the deletion of the autophagy-related protein Atg16L1 induced caspase-1 activation, which led to an enhanced pyroptosis and increased IL-1β release [60]. Studies also found that, in the process of atherosclerosis, autophagy deficiency may promote atherosclerosis by overactivating the inflammasome [61]. Similarly, Pu et al. found that Atg7 deficiency in pseudomonas aeruginosa sepsis led to inflammasome activation, accompanied by an enhanced macrophage pyroptosis and the increased release of inflammatory factors [62]. Based on these findings, the researchers hypothesized that autophagy may negatively regulate pyroptosis. In recent years, there have been many reports on the relationship between inflammasome-mediated pyroptosis and autophagy. For example, the inhibition of autophagy can promote caspase-1and NLRP3 inflammasome activation, leading to an increased IL-1β and IL-18 secretion [63]. Autophagy protects infected macrophages and microglia from pyroptosis [64,65]. These results suggested that the inhibition of autophagy could promote inflammasome-mediated pyroptosis.

Autophagy also plays a regulatory role in GSDME-mediated pyroptosis, but its mechanism remains unclear. Yu et al. demonstrated the relationship between autophagy and GSDME-mediated pyroptosis for the first time, and found that the inhibition of autophagy enhanced DOX-induced pyroptosis [66]. However, the mechanism of the interaction between autophagy and the GSDME-mediated pyroptosis was not revealed, suggesting that the AMPK-EEF-2K signaling pathway plays an important role in this process.

In conclusion, autophagy may be a negative regulator of pyroptosis (Figure 7B). This seems to be a promising way to promote pyroptosis, i.e., by inhibiting autophagy. However, it is debatable whether autophagy contributes to pyroptosis under any circumstances. In addition, the specific mechanism by which autophagy inhibits pyroptosis remains unclear. Therefore, determining how autophagy exerts a general inhibitory effect on GSDME-mediated pyroptosis is an important question in the field.

#### 4.2.3. Promoting ROS Production

Reactive oxygen species (ROS) are closely related to the occurrence of tumors. However, excessive ROS can cause apoptosis or necrosis by enhancing cellular oxidative stress. Therefore, increasing ROS levels in tumor cells through the use of chemotherapy drugs has been a widely used strategy. Recently, it was found that ROS can induce pyroptosis by activating the nucleotide-binding domain and leucine-rich repeat protein-3 (NLRP3) inflammasome (Figure 7C) [67]. For example, Xu et al. coated piperlongumine (PL) with iron-containing MOFs, which could trigger the Fenton reaction, increase the ROS, and, finally, cause the ferroptosis and pyroptosis of tumor cells [68]. In addition, Nadeem et al. synthesized a virus-spike tumor-activatable pyroptotic agent (VTPA) with an organosilicon-coated iron oxide nanoparticle core and spiky manganese dioxide protrusion. A high concentration of glutathione (GSH) in the tumor microenvironment could effectively degrade the organosilicon hybrid shell containing disulfide bonds. As a result, the Mn and Fe ions were released to elicit the Fenton-like reaction for the rapid ROS generation, which synergistically activated the NLRP3 inflammasomes to induce cancer cell pyroptosis [69].

In addition to activating the NLRP3 inflammasome to induce cancer cell pyroptosis, ROS can also induce pyroptosis through the caspase-3/GSDME pathway. For example, Zhou et al. cotreated melanoma cells with ferric ion and carbonyl cyanide m-chlorophenizone (CCCP), which significantly increased the ROS level and could activate caspase-3 to mediate GSDME lysis and trigger pyroptosis [70]. In addition, studies have shown that photodynamic therapy (PDT) can increase the production of intratumoral ROS and promote pyroptosis [71,72]. In addition, in order to reduce the damage of drugs to normal tissues, Xiao et al. constructed a nanodrug delivery system to codeliver chemotherapy drugs and photosensitizers to tumor sites. After laser irradiation, ROS in tumors increased, thereby inducing the pyroptosis of tumor cells [73]. These studies suggest that inducing an ROS storm in the tumor microenvironment may be an effective cancer treatment method.

#### 4.2.4. The Delivery of GSDM Protein

The main effector proteins of pyroptosis are GSDM family proteins with membrane pore-forming activity; thus, directly delivering GSDM family proteins could also induce pyroptosis (Figure 7D). Shao et al. combined active GSDMA3 protein with gold nanoparticles using a silane linker, which could be decomposed through phenylalanine trifluoroborate (phi-bf3) desilication; thus, nanoparticles and GSDMA3-N can be selectively accumulated in tumors [53]. Given that the GSDM family proteins are the direct-acting molecules in inducing tumor pyroptosis, the construction of nanodelivery systems for the delivery of GSDM family proteins is likely to become a therapeutic hotspot.

## 5. Blockage Pyroptosis to Decrease Side Effects

Taking specific means to activate GSDME at tumor sites to enhance immune efficacy is a good antitumor strategy. Unfortunately, studies have shown that the expression of GSDME is more abundant in normal tissues, such as the brain, heart, and kidneys, than in tumor cells; thus, these drugs can cause harm to normal tissues (Figure 8). In this section, we aimed to introduce pyroptosis-induced body damages and diseases, and provide methods to block pyroptosis to decrease potential side effects.

### 5.1. GSDME-Mediated Tissue Damages and Disease Development

#### 5.1.1. Tissue Damages

Based on the fact that most drugs lack specific targeting and high GSDME expression in normal tissues, toxicity and side effects are often caused during the treatment process. For example, cisplatin has been shown to have an extraordinary effect in the treatment of cancer, but it has been clinically found to be prone to producing a variety of toxicity and side effects during treatment, such as nephrotoxicity, ototoxicity, and bone marrow suppression [74,75,76,77]. Among them, nephrotoxicity is a dose-limiting toxicity of cisplatin, which severely limits the application of the drug in the clinic (Figure 8A). Therefore, reducing its side effects during treatment has been a meaningful direction for researchers. Recent studies have reported that cisplatin-induced nephrotoxicity is associated with the excessive activation of GSDME in renal tubular cells, and this study brought light to scientists [78]. In addition to cisplatin, another chemotherapeutic agent, doxorubicin, has also been impeded in clinical applications by severe cardiotoxicity during treatment. Recently, researchers found that the excessive pyroptosis of cardiomyocytes may be associated with their mediated cardiotoxicity [79] (Figure 8A). Moreover, studies have reported that the excessive activation of GSDME can also affect the function of related reproductive organs, such as causing testicular damage [80] (Figure 8A).

#### 5.1.2. Autoimmune Diseases

The immune system is comprised of immune cells, immune tissues, and immune organs, which can protect the body from foreign invading substances such as bacteria, parasites, viruses, and cancer cells. However, the abnormal response of the acquired immune system may result in autoimmune diseases (ADs). ADs are a class of diseases where immune cells target their own healthy tissues by mistake and release signals to prompt the body to attack them. In recent years, the incidence of ADs has increased, having a huge impact on the patients themselves and their social economy. Recent studies have reported that the activation of GSDME-related pyroptosis can induce autoimmune-related diseases, such as inflammatory factor release syndrome. As is well-known, CAR-T therapy is often used to treat leukemia and lymphoma, but there are obvious adverse effects to this method, namely, inflammatory factor storm. A recent study found that the side effect mechanism lies in the fact that CAR-T therapy releases large amounts of active perforin 1 (PRF1) and granzyme B(GZMB), which overactivate the caspase-3/GSDME pathway and lead to pyroptosis. At the same time, DMAPs, such as adenosine triphosphate (ATP) and HMGB1, are released, which stimulate macrophages to release proinflammatory factors and cause a severe inflammatory factor storm [81]. Rheumatoid arthritis is another type of highly prevalent autoimmune disease, which can seriously reduce the quality of life for patients (Figure 8B). However, the cause of the disease is not completely clear at present. Recent studies reported that there are abundant necrosis factors (TNFs) in the synovial fluid and tissues of these patients [82]. As we all know, TNFs are strong inducers of apoptosis, which can activate caspase-3 to cleave GSDME that then release a lot of inflammatory cytokines and further deteriorate the disease. Therefore, GSDME-targeting treatments may offer hope for patients with autoimmune-related diseases.

#### 5.1.3. Inflammatory Diseases

Pyroptosis is regarded as the first line of defense against danger. When cells are infected by pathogens, they often cause cell damage and release proinflammatory mediators, including IL-1β and HMGB1, to recruit immune cells, including macrophages. Pyroptosis can protect the body from infection and injury to a certain extent, but the excessive activation of this pathway often causes body damage, such as the induction of inflammatory diseases. In other word, the accumulation of numerous inflammatory cytokines could induce certain diseases, causing serious damage to the body, such as sepsis, pneumonia [83], Crohn’s disease [84], etc. (Figure 8C). Sepsis is the most classic example of a pyroptosis-related inflammatory disease. It is a life-threatening disease caused by the dysregulation of the body’s response to infection, which leads to multiorgan dysfunction [85]. The pathology of sepsis is complex, presenting with rapid progress and high mortality. There has been no effective drug treatment developed thus far, resulting in clinics mainly relying on antibiotics and organ function support, but of which the cost is high [86,87]. Even after a successful treatment response, there are still many life-long complications of the disease, which seriously affect the quality of life. Based on this, the search for effective drugs to treat sepsis is of great significance. A recent study mentioned that the excessive activation of pyroptosis is an important cause of the immune dysregulation of sepsis and an increased risk of death [88]. Studies have shown that sepsis is associated with GSDMD, but whether it is related to the activation of GSDME remains to be elucidated [89]. Therefore, an in-depth elucidation of the pathophysiological mechanisms between pyroptosis and sepsis could bring new light to the treatment of sepsis.

### 5.2. Methods to Block Pyroptosis

#### 5.2.1. Reducing the Expression of Full-Length GSDME Protein

Some researchers have reported that the protein expression of GSDME in normal tissues is much higher than in tumor tissues, which may be related to the methylation of the GSDME gene promoter in tumor cells. Regardless of what the reason is, the high expression of GSDME in normal tissues brings serious toxic effects to the organism. Protein is the executor of life activities, and protein expression is often affected by transcription, translation, and post-translational modification processes. Therefore, in order to prevent GSDME from inducing pyroptosis in normal tissues, it is possible to affect its function by blocking the above steps from occurring. For example, gene interference, the promotion of post-translational modifications, and other means can be used to affect its function (Figure 9A).

Gene interference refers to the inhibition of the expression of genes of interest in a specific way, so that they cannot be expressed to achieve the purpose of treating diseases, and mainly includes RNA (RNAi), CRISPR-Cas 9, and other interferences [82,90]. Through these techniques, researchers identified that the executors of pyroptosis are GSDMD and GSDME, and were able to figured out the related pathways and biochemical functions of GSDME [91,92]. The post-translational modification, on the other hand, adds groups to specific amino acid branch chains or chemically modify the existed groups [93]. Post-translational modifications include phosphorylation, methylation, acetylation, palmitoylation, etc., which can directly regulate the activity of proteins and expand their chemical structure functions, so as to increase the complexity and diversity of the proteins. Currently, there have been few reports on the post-translational modification of the gasdermin family. For instance, Hu et al. noted that palmitoylation was closely related to GSDME-induced pyroptosis. Their studies showed that palmitoylation inhibitors could inhibit the separation of between the active N-terminus and C-terminus, playing an important role in inhibiting cell pyroptosis [94]. In addition, it was also reported that when GSDME is phosphorylated, it also inhibits its activity, which may be associated with the suppression of the GSDME-N oligomerization [38]. However, the authoritative mechanism still remains to be explored. Therefore, it is a simple but effective way to focus on the GSDME protein expression process and reduce its expression.

#### 5.2.2. Blocking GSDME Cleavage

Even though full-length GSDME protein are abundantly expressed in normal cells, GSDME-C and the GSDME-N-terminus are often tightly connected, resulting in an autoinhibited state. Their activity depends on the breakage of the hinge region. Therefore, the inhibition of the GSDME cleavage can also block pyroptosis (Figure 9B).

As shown above, the cleavage of GSDME often requires the participation of caspase-3 or granzyme B, so inhibiting the activation of caspase-3 or the release of granzyme B are effective ways to block pyroptosis [95]. For example, Xu et al. synthesized two novel cell-permeable inhibitors, Ac-DMPD-CMK and Ac-DMLD-CMK, which were found to possess the function of inhibiting the cleavage activity of caspas-3, thereby reducing the liver tissue toxicity [95].

It has been shown that upstream caspase-3 can only recognize “DMPD”—the cleavage site of GSDME. If this sequence were to be mutated or modified, it would affect the occurrence of pyroptosis [11]. For example, Shao Feng et al. conversed the caspase1 cleavage site -FLTD- to caspase-3 cleavage site -DEVD-, which can lead to pyroptosis rather than apoptosis in cells with a high expression of GSDME protein [11]. In addition, some researchers changed the expression sequence of GSDME to inhibit the activation of GSDME through bio-orthogonal reactions. Bio-orthogonal reactions refer to a type of chemical reaction that can be carried out in biological systems without interfering with natural biochemical processes. For instance, researchers have developed BaseBAC (a chemically controllable C-to-T base editing technology in living cells) to regulate the autoinhibitory C-terminal expression of GSDME protein (base editing of CAA to TAA at Q287) by introducing stop codons in the open reading frame, causing GSDME to terminate the translation and only express the active N-terminus, so as to chemically regulate pyroptosis [47,96]. Although this study reported that orthogonal reactions were used to induce pyroptotic cell death rather than inhibition, through the in-depth understanding of the orthogonal response, the purpose of inhibiting pyroptosis could also be achieved by terminating the translation early of GSDME-N, or repeating the expression of GSDME-C. Therefore, the modification of the GSDME sequence through specific means is also a promising therapeutic tool.

#### 5.2.3. Inhibiting Pore-Forming Activity of GSDME-N

The activity of the GSDME-N-terminal lies in that it can bind to phospholipids on the membrane, participate in oligomerization, and form pores 10–15 nm in size, allowing for a small volume of inflammatory factors and hypotonic fluid to pass freely to cause pyroptosis. Based on the above theory, blocking its binding to phospholipids on the membrane or oligomerization can inhibit pyroptosis (Figure 9C).

How can the binding of GSDME-N to phospholipids on the membrane be inhibited? Some studies have shown that GSDME-N is selective of membrane lipids, and it might only bind to lipids in the membrane such as PI, and P2 not out of the membrane [11]. However, no relevant reports have reported whether changing the type or location of the phospholipids on the membrane can inhibit pyroptosis. It may be because phospholipids play important functions on cell membranes, and whether this is a viable strategy remains to be studied.

GSDME-N often requires multiple oligomerizations to form larger pores to allow for the leakage of the contents and the induction of pyroptosis. Based on this, some researchers screened the phosphorylation sites of GSDME with proteomic mass spectrometry, and found that its activity was inhibited by phosphorylation [42]. In order to explore the mechanism of its inhibition, confocal microscopy and oligomerization detection methods were selected to track GSDME-related activities. The results showed that GSDME’s T6A was phosphorylated and exerted its activity through inhibiting its oligomerization on the membrane rather than its targeting to the membrane. In 2020, a published science article mentioned that GSDME can be modified using fumaric acid to add a 2-(succinyl), thereby inhibiting its formation of oligomers and blocking pyroptosis [97]. In addition, our research group found that poly-EGCG nanoparticles have the ability to inhibit the oligomerization of GSDMD, but whether this can also affect the related functions of GSDME remains to be explored [98]. Based on the above analysis, inhibiting the oligomerization of GSDME-N on the membrane can inhibit pyroptosis, while the underlying mechanism waits to be discovered.

## 6. Challenges in Balancing Cell Apoptosis and Pyroptosis

Cell death is an extremely complex process, of which there are many forms. In some cases, tumor cells are killed through different ways, such as apoptosis, pyroptosis, and autophagy. For a long time, researchers believed that the treatment and prognosis of cancer were closely related to apoptosis. However, an important reason for the resistance of tumor cells to drugs is the gradually increased antiapoptosis ability of tumor cells. Pyroptosis is a type of death that can cause inflammatory cell death. It improves the tumor immune microenvironment by releasing inflammatory factors, thereby activating an antitumor immune response. Thus, the transition from apoptosis to pyroptosis can bypass antiapoptotic barriers and overcome chemical resistance. In addition, the pyroptosis of normal cells mediated through chemotherapy drugs increases the toxicity and side effects. Therefore, methods for how to balance apoptosis and pyroptosis are a major problem in cancer treatment. Recently, researchers found that caspase-3 can link apoptosis with pyroptosis. However, the different expression of GSDME in different cells determined the death induced by caspase-3 activated through the use of chemotherapy drugs. In addition, the biodistribution of drugs is the other important issue determining the therapeutic index. Therefore, the main contradictory points of apoptosis and pyroptosis currently lie in the differential expression of GSDME and the targeting of drugs.

### 6.1. GSDME Expression

When caspase-3 is activated by chemotherapeutic agents, the expression level of intracellular GSDME determines the different modes of cell death. When GSDME was highly expressed, caspase-3 cleaved GSDME to mediate pyroptosis; on the contrary, it mediated apoptosis. However, due to the methylation of the GSDME promoter region in most tumor tissues, GSDME expression tends to be lower than that in normal tissues [54,55,99], making it difficult for chemotherapy drugs to activate tumor cell pyroptosis to exert a therapeutic effect, also increasing the side effects of chemotherapy. Based on this, researchers found that DNA methyltransferase inhibitors could inhibit the hypermethylation of the GSDME promoter region in tumor cells with a low GSDME expression, thereby increasing the expression of GSDME in tumor cells and inducing pyroptosis. In addition, the expression of GSDME in normal tissues enables chemotherapeutic drugs to activate GSDME through caspase-3 cleavage and induce the pyroptosis of normal cells, which leads to different degrees of damage in normal tissues. Therefore, ways to balance the expression of intracellular GSDME are a big problem.

### 6.2. Targeting Drug Delivery

Since the concept of pyroptosis was first proposed in 2001, research on tumor cell pyroptosis induced through chemotherapeutic drugs has seen revolutionary progress. However, a series of obstacles still needs to be overcome before the drug reaches the tumor site, including the insolubility of the drug, short half-life, low bioavailability, and nonspecific biodistribution. Therefore, the accurate delivery of drugs to the tumor is crucial. In the past decade, with the development of biomaterials and nanotechnology, more and more studies have used nanodelivery systems (NDSs) to reduce drug toxicity, improve drug bioavailability, and achieve efficient drug accumulation at tumor sites through targeted modification [100,101]. At present, the NDS carriers used to load tumor pyroptosis drugs mainly include liposomes, hydrogels, polymer micelles, metal–organic frameworks (MOFs), and biomimetic cell membranes. For example, Fan et al. developed a cisplatin-loaded nanoliposome (LipoDDP) that significantly increased the drug loading rate of cisplatin and alleviated its toxicity to normal cells. When LipoDDP is combined with the DNA methyltransferase inhibitor DAC, it can synergically induce the pyroptosis of colon cancer cells by restoring the expression of GSDME protein [15]. Zhao et al. constructed a biomimetic nanoparticle (BNP) wrapped in a breast cancer cell membrane loaded with DAC and indocyanine green (ICG); the results showed that BNP endocytosis was stronger than free ICG and IDNP (without membrane encapsulation) in 4T1 cells, and showed a stronger pyroptosis induction effect [102].

The advantages of NDS make the drug-targeted induction of tumor cell pyroptosis more feasible, so combining chemotherapeutic drugs with NDSs is a very promising cancer treatment strategy. The strategy of drug-induced tumor cell pyroptosis based on NDSs can overcome the shortcomings of using small molecule pyroptosis inducers alone, such as rapid clearance in vivo, easy to cause systemic adverse reactions and weak tumor-targeting ability. Although it has achieved good results in antitumor experiments, there are still many problems to be explored in this strategy. For example, more than 50% of the antitumor nanomaterials currently used in clinical application are liposomes, but the low encapsulation efficiency of liposomes has always been an unavoidable problem. In addition, the vast majority of NDSs are in the experimental stage with less clinical translations. Therefore, how to achieve the accurate and efficient targeting of chemotherapy drugs to tumor cells is still a difficult problem.

## 7. Conclusion and Perspectives

Chemotherapeutics is an important modality for cancer treatment, and both apoptosis and pyroptosis are important pathways for the efficacy. Caspase-3 is an important upstream regulator of both. It cannot only activate APF-1 to induce apoptosis, but also shear GSDME to induce pyroptosis, and which way cells go depends on how much the executing molecule GSDME is expressed. Based on the debates that pyroptosis at tumor sites improves immune efficacy and pyroptosis in normal tissues induce toxic side effects, this review systematically summarized the key mechanisms of chemotherapeutic drug-induced apoptosis and pyroptosis, and the methods of pyroptosis and apoptotic transformation in tumor and normal tissues from the mechanism and pharmacological fields, which intends to provide more insights into the relevant applications of chemotherapeutic drugs.

The study of GSDME not only fills in the insights of cell pyroptosis, but also promotes the development of preventive treatments for chemotherapeutic drug resistance and toxic side effects. Research on GSDME is in full swing, but basic research is still relatively limited, and there is still a broad space to be explored. Below are some areas where existing research has limitations, but could still be explored in-depth: 1. Investigators focused only on the single effect of GSDME, so the efficacy and toxic side effects could not be taken into account at the same time, and the drug treatment was often associated with both. How to balance the relationship between the two is a very important research point. Nanotechnology is widely used by researchers in cancer treatments, so can this technique combined with pyroptosis provide an effective therapeutic direction for this? 2. Researchers only focused on the effects of GSDME in tumor cells and normal tissues, but there are few reports on the expression of GSDME in immune cells. Pyroptosis can enhance the body’s immune response in a variety of ways, thus, playing a role in antagonizing infection and endogenous danger signals. Focusing on the expression and related efficacy of GSDME in immune cells may be closely related to disease-related progression. 3. The existing means of regulating the expression level of GSDME are limited, and other disciplines can be integrated to explore the study of pyroptosis.

## Figures and Tables

**Figure 1 cancers-15-00026-f001:**
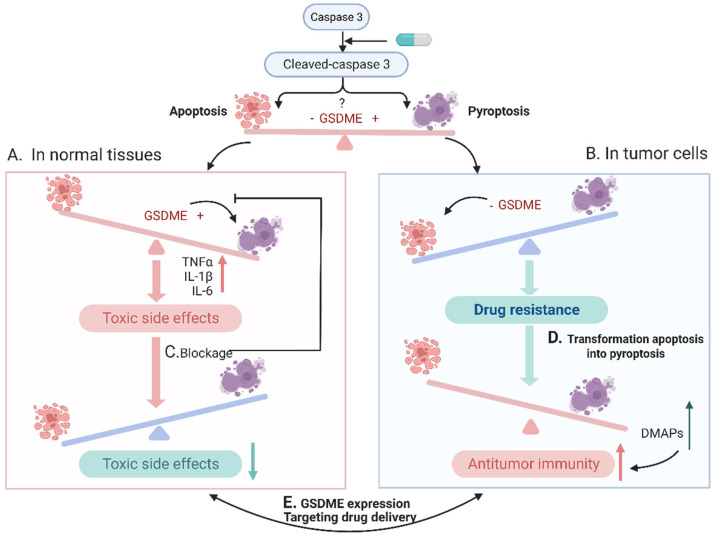
A brief overview of the main contents in this review: caspase-3 can be converted to cleaved caspase-3 after activation through specific pathways, such as chemotherapeutic agents, and this active molecule can activate downstream apoptosis and pyroptosis, and the occurrence of apoptosis or pyroptosis depends mainly on the expression level of GSDME. (**A**) In normal tissues, the expression level of GSDME tends to be high, and when activated with chemotherapeutic drugs, the cells undergo pyroptosis and produce serious toxic side effects. (**B**) In tumor cells, GSDME is methylated at low levels and apoptosis occurs, while tumor cells tend to resist apoptosis through different pathways, resulting in drug resistance and reduced antitumor efficacy. (**C**) The means to block the onset of pyroptosis in normal cells may reduce the toxic effects. (**D**) Methods are taken in tumor cells to convert apoptosis into pyroptosis, which can release DMAPs and other immunogenic substances to activate tumor immunity and improve antitumor efficacy. (**E**) The main point of how to achieve a precise balance between pyroptosis and apoptosis via modulating GSDME expression and targeting drug delivery.

**Figure 2 cancers-15-00026-f002:**
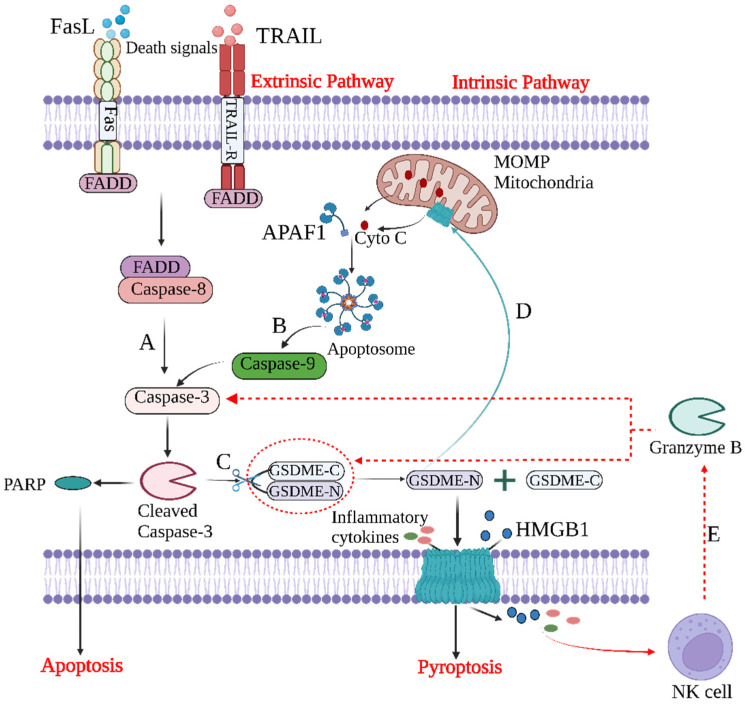
The crosstalk between apoptosis and GSDME-dependent pyroptosis. (**A**) The extrinsic pathway and (**B**) intrinsic pathway of apoptosis. (**C**) GSDME-mediated pyroptosis. After caspase-3 cleavage, GSDME is activated to produce GSDME-N, which is transferred to the plasma membrane and perforated, releasing inflammatory cytokines and HMGB1. (**D**) GSDME-N targets mitochondrial membranes to form pores, leading to the release of cytochrome C and the activation of caspase-3, enhancing pyroptosis. (**E**) Granzyme B released by NK cells can, on the one hand, activate caspase-3, and, on the other hand, directly cleave GSDME to cause pyroptosis.

**Figure 3 cancers-15-00026-f003:**
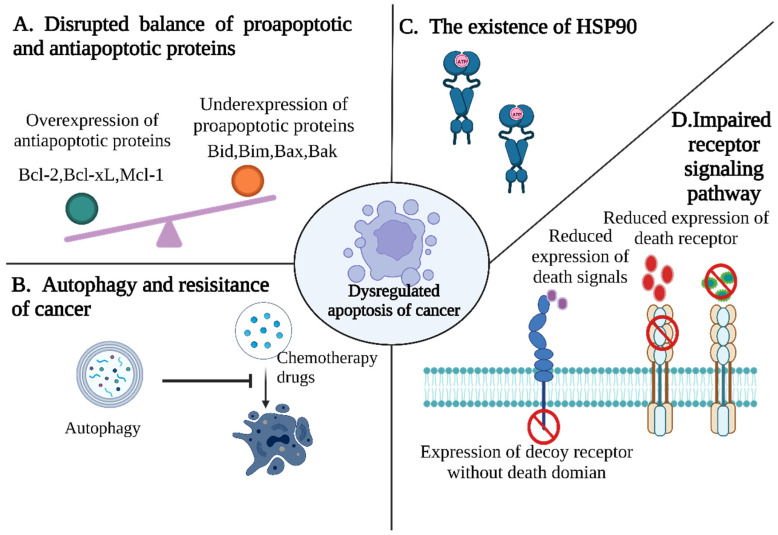
Pathways contributing to evasion of apoptosis in tumor cells. (**A**) Disrupting balance of proapoptotic and antiapoptotic proteins. (**B**) Autophagy inhibits the efficacy of various anticancer drugs and protects cancer cells from apoptosis. (**C**) Hsp90 acts to resist chemotherapy-induced apoptosis by assisting in the folding, stabilization, and maturation of different types of oncoproteins. (**D**) Impaired death receptor signaling.

**Figure 4 cancers-15-00026-f004:**
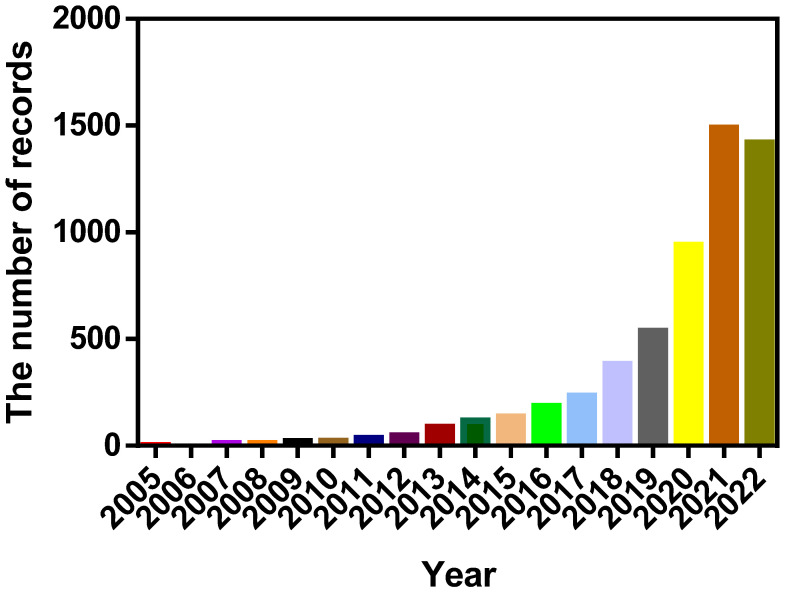
Reports involving pyroptosis increased over the past 18 years. The records were obtained from PubMed.

**Figure 5 cancers-15-00026-f005:**
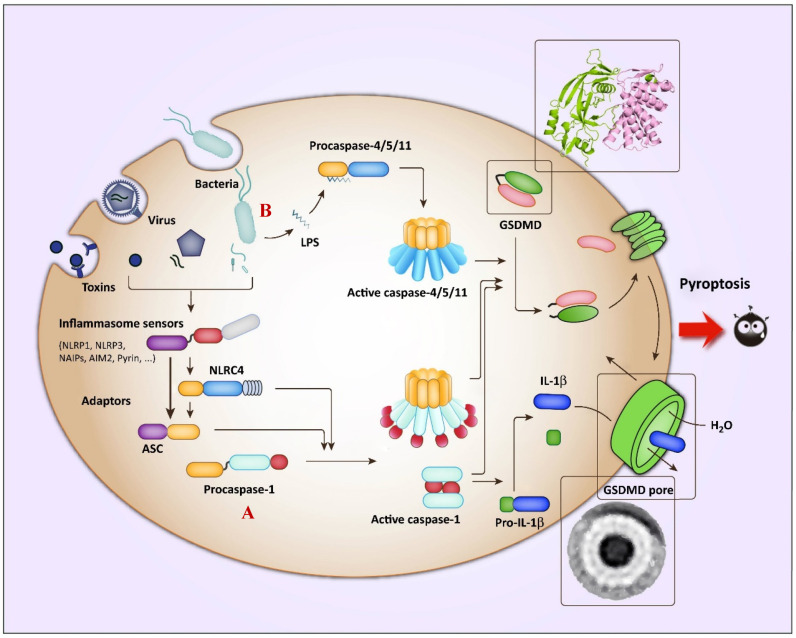
Overview of the pathways involved in GSDMD-induced pyroptosis. (**A**) The canonical inflammasome pathways: viruses and toxins activate various inflammasomes, thereby activating caspase-1 to mediate the maturation of proinflammatory cytokines (e.g., IL-1β and IL-18) and cleave GSDMD to induce pyroptosis. (**B**) The noncanonical inflammasome pathways: the LPS in the cell wall of the invading bacteria mediates the activation of murine caspase-11 or human caspase-4/5, and similarly cleaves GSDMD to induce pyroptosis. Copied with permission [10]. Copyright 2017, *Trends in Biochemical Sciences*.

**Figure 6 cancers-15-00026-f006:**
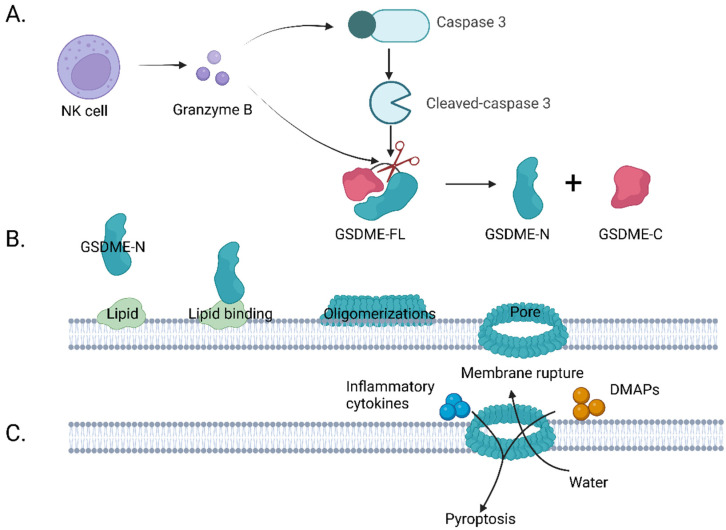
Schematic illustration of the GSDME activation steps to induce pyroptosis. (**A**) GSDME-N-terminal fragment liberate: the GSDME full-length protein can be cleaved with cleaved caspase-3 or granzyme B to release the GSDME-N terminus. (**B**) The GSDME-N-terminus binds to the phospholipids in the membrane through an electrostatic interaction and oligomerizes into pores in the membrane. (**C**) Cell membrane swelling and rupture: inflammatory factors and immunogenic substances are released through the pores to outside the membrane to form an inflammatory environment. In addition, water outside the membrane enters the cell due to an osmotic pressure difference, and the cell membrane swells and ruptures.

**Figure 7 cancers-15-00026-f007:**
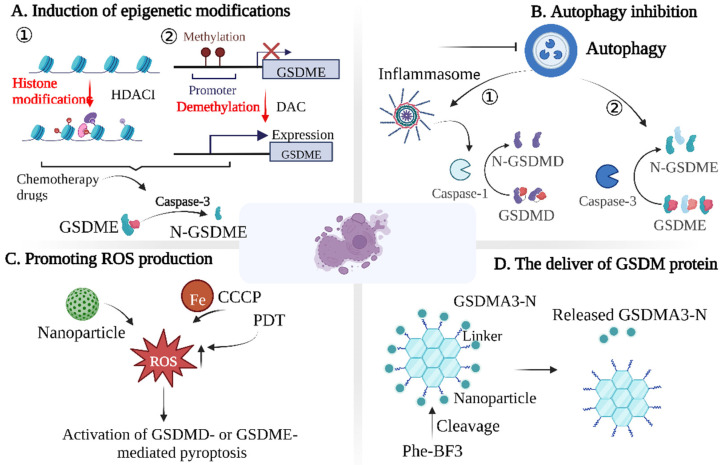
Strategies to promote pyroptosis in tumor cells. Induced pyroptosis can be divided into four dimensions: (**A**) induction of epigenetic modifications; (**B**) autophagy inhibition; (**C**) promoting ROS production; (**D**) the delivery of GSDM protein.

**Figure 8 cancers-15-00026-f008:**
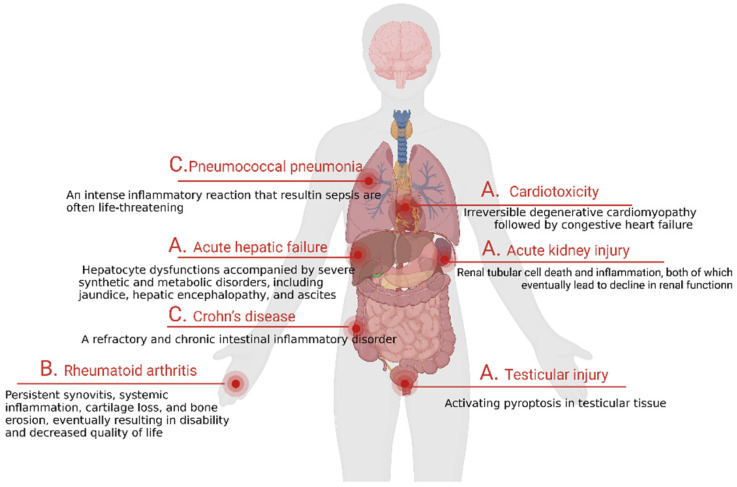
GSDME contributes to the toxicity of chemotherapy. Chemotherapeutic drugs can induce pyroptosis in normal tissues, resulting in tissue damage and a cytokine storm. The toxicity and side effects mainly include (**A**) tissue damage, (**B**) autoimmune diseases, and (**C**) inflammatory diseases.

**Figure 9 cancers-15-00026-f009:**
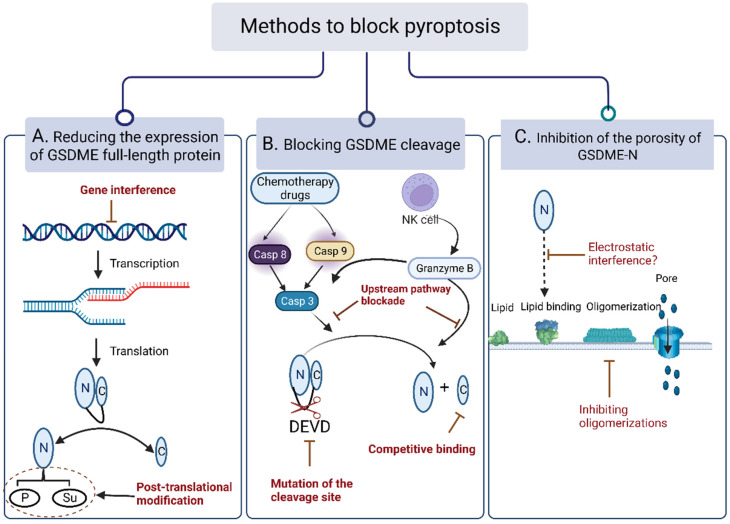
Blocking the pyroptosis pathway in normal cells. (**A**) Reducing the expression of full-length GSDME protein via gene interference and post-translational modification. (**B**) Blocking GSDME cleavage through hinge region breakage, mutation of the cleavage site, and competitive binding of GSDME-C-terminus. (**C**) Inhibition of the porosity of GSDME-N through electrostatic interference and inhibiting oligomerizations, but whether electrostatic interference is effective remains to be studied.

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
