# Peer review of "Balance Cell Apoptosis and Pyroptosis of Caspase-3-Activating Chemotherapy for Better Antitumor Therapy"

_cancers, 2022, doi:10.3390/cancers15010026_

Round 1

Reviewer 1 Report

This manuscript comprehensively reviews the strategy by controlling the balance between pyroptosis and apoptosis in tumor and normal cells to enhance cancer therapy efficiency and reduce the side effects to the patients. This is an interesting review but needs to correct some queries as followed:

1. Some abbreviation names in the text should give the full names when they are used at first time, such as GSDME, GSDMD, HMGB1 and DMAP in the page 2.

2. Please give an abbreviation list for all of the abbreviated names.

3. At the line 125 of page 3, an addition space between “inhibit apoptosis” should be deleted.

Author Response

Reviewer #1

This manuscript comprehensively reviews the strategy by controlling the balance between pyroptosis and apoptosis in tumor and normal cells to enhance cancer therapy efficiency and reduce the side effects to the patients. This is an interesting review but needs to correct some queries as followed:

  1. Some abbreviation names in the text should give the full names when they are used at first time, such as GSDME, GSDMD, HMGB1 and DMAP in the page 2.

Author reply: Thanks for pointing this out, and we have revised the manuscript accordingly.

  1. Please give an abbreviation list for all of the abbreviated names.

Author reply: Thanks for your valuable comments. We have organized all the abbreviations into a table placed in front of the references.

  1. At the line 125 of page 3, an addition space between “inhibit apoptosis” should be deleted.

Author reply: Thank you for pointing this out, and this has been revised.

Reviewer 2 Report

The review manuscript by Lingjiao and colleagues is a really interesting recollection of the literature on the role of gasdermins in anti-tumor immunity. It is easy to read and the figures are very self-explanatory. I enjoyed the reading and suggest a minor revision of the manuscript that may help improve the narration and accuracy of some statements.

Page 2, line 46: it is unclear what the authors intend to say by “inflammatory vesicle-caspase1-GSDMD….”. Is this the classical inflammasome? Why do they refer to this as a “vesicle”?

Page 2, line 61: the term “DMAP” is unclear (also in figure 6)

Section 2 (lines 129-133) and Figure 2: Whereas Fas and TRAIL-Rs form a cell death inducing platform (DISC), this is not the case for TNFR1. At least not as primary signalling output. Even though what the authors claim about extrinsic cell death in general and the Fas/CD95 recruits Casp8, this is not entirely correct for TNFR1. Since I understand that this is not the main focus of the review and the authors remain general and vague, I would suggest that the authors only mention Fas/CD95 (and or TRAIl/TRAILRs) and avoid mentioning TNFR1. Otherwise, the authors should clarify that cell death is not the primary output of TNR1-signalling.

Page 6, lines 196-199: please revise this sentence since it is not understandable.

Page 8, line 249: “…GSDMD has promoted the development of pyroptosis…”, do the authors mean “inhibitors”? otherwise, please revise this sentence as it is not clear.

Section 6 and 6.1 is quite repetitive to previous concepts discussed in the manuscript. The authors should try to unify the concepts and stream line section 6.

Please revise the text for grammar and typos

Author Response

Reviewer #2

The review manuscript by Lingjiao and colleagues is a really interesting recollection of the literature on the role of gasdermins in anti-tumor immunity. It is easy to read and the figures are very self-explanatory. I enjoyed the reading and suggest a minor revision of the manuscript that may help improve the narration and accuracy of some statements.

  1. Page 2, line 46: it is unclear what the authors intend to say by “inflammatory vesicle-caspase1-GSDMD….”. Is this the classical inflammasome? Why do they refer to this as a “vesicle”?

Author reply: Thanks for your comment. The inflammatory vesicle mentioned in the manuscript refers to the classical inflammasome, and it is not a “vesicle” but a protein complex consisting of three main components, i.e., the receptor protein (NLRP1, NLRP3, NLRC4, AIM2), the junction protein ASC, and the downstream cystatinase caspase-1. To follow your suggestion, we have replaced “inflammatory vesicle” by the more professional statement “the classical inflammasome”.

  1. Page 2, line 61: the term “DMAP” is unclear (also in figure 6)

Author reply: Thanks for pointing this out. DMAP is the abbreviation of damage associated molecular patterns, which refers to endogenous molecules released from death cell, mainly including calreticulin, high mobility group protein, ATP, heat shock proteins. We have made the explanation of “DMAP” in revised manuscript to address the Reviewer’s comment.

  1. Section 2 (lines 129-133) and Figure 2: Whereas Fas and TRAIL-Rs form a cell death inducing platform (DISC), this is not the case for TNFR1. At least not as primary signalling output. Even though what the authors claim about extrinsic cell death in general and the Fas/CD95 recruits Casp8, this is not entirely correct for TNFR1. Since I understand that this is not the main focus of the review and the authors remain general and vague, I would suggest that the authors only mention Fas/CD95 (and or TRAIl/TRAILRs) and avoid mentioning TNFR1. Otherwise, the authors should clarify that cell death is not the primary output of TNR1-signalling.

Author reply: Thank you for your comments. We have removed TNFR1 and changed it to “The extrinsic pathway is initiated by members of the death receptor family, such as Fas (CD95) receptor, TRAIL-R1 (DR4), and TRAIL-R2 (DR5), which bind to their corresponding ligands. Fas-associated death domain (FADD) and cysteine proteases such as caspase-8 were recruited to form death-inducing signaling complex (DISC). This complex can activate downstream caspase-3, which eventually leads to apoptosis”. Corresponding modifications have also been made in Figure 2.

  1. Page 6, lines 196-199: please revise this sentence since it is not understandable.

Author reply: Thanks for pointing this out, and we have changed the statement by wording “For example, the leukemia 2 (bcl-2) family are among the earliest genes associated with apoptosis that have been studied. Bcl-2 plays a key role in the regulation of apoptosis and is composed of pro-apoptotic and anti-apoptotic proteins. When the balance between them is disrupted, it leads to an imbalance of apoptosis”.

  1. Page 8, line 249: “…GSDMD has promoted the development of pyroptosis…”, do the authors mean “inhibitors”? otherwise, please revise this sentence as it is not clear.

Author reply: Thanks for your comment. This sentence wanted to emphasize that the identification of GSDMD structure is helpful for better understanding the molecular mechanisms of GSDMD-mediated pyroptosis. To follow the reviewer’s suggestion, we have revised the statement by wording “The identification the structure and physiological function of GSDMD would provide researchers with a deeper understanding of pyroptosis and a more promising direction for the treatment of cancer and related diseases.”

  1. Section 6 and 6.1 is quite repetitive to previous concepts discussed in the manuscript. The authors should try to unify the concepts and stream line section 6.

Author reply: We thank the Reviewer for this suggestion. Section 6 is a summary and Section 6.1 is a detailed discussion. We have deleted the repetition to make it more concise.

  1. Please revise the text for grammar and typos.

Author reply: Thanks for your suggestion, and we have performed a careful proof-reading to revise the English.